# Metabolism of *cis*- and *trans*-Resveratrol and Dihydroresveratrol in an Intestinal Epithelial Model

**DOI:** 10.3390/nu12030595

**Published:** 2020-02-25

**Authors:** Veronika Jarosova, Ondrej Vesely, Ivo Doskocil, Katerina Tomisova, Petr Marsik, Jose D. Jaimes, Karel Smejkal, Pavel Kloucek, Jaroslav Havlik

**Affiliations:** 1Department of Food Science, The Faculty of Agrobiology, Food and Natural Resources, The Czech University of Life Sciences Prague, 16500 Prague, Czech Republic; jarosovav@af.czu.cz (V.J.); veselyo@af.czu.cz (O.V.); tomisova@af.czu.cz (K.T.); marsik@af.czu.cz (P.M.); kloucek@af.czu.cz (P.K.); 2Department of Microbiology, Nutrition and Dietetics, The Faculty of Agrobiology, Food and Natural Resources, Czech University of Life Sciences Prague, 16500 Prague, Czech Republic; doskocil@af.czu.cz; 3Department of Natural Drugs, Faculty of Pharmacy, University of Veterinary and Pharmaceutical Sciences Brno, 61242 Brno, Czech Republic; karel.mejkal@post.cz

**Keywords:** Caco-2 cell lines, glucuronidation, phenolics, stilbenoids, sulphatation, UHPLC-MS-Q-TOF

## Abstract

*Trans*-resveratrol, a well-known plant phenolic compound, has been intensively investigated due to its association with the so-called French paradox. However, despite its high pharmacological potential, *trans*-resveratrol has shown relatively low bioavailability. *Trans*-resveratrol is intensively metabolized in the intestine and liver, yielding metabolites that may be responsible for its high bioactivity. The aim of this study was to investigate and compare the metabolism of *trans*-resveratrol (tRes), *cis*-resveratrol (cRes) and dihydroresveratrol (dhRes) in an in vitro epithelial model using Caco-2 cell lines. Obtained metabolites of tRes, cRes and dhRes were analyzed by LC/MS Q-TOF, and significant differences in the metabolism of each compound were observed. The majority of tRes was transported unchanged through the Caco-2 cells, while cRes was mostly metabolized. The main metabolite of both *cis-* and *trans-*resveratrol observed as a result of colon microbial metabolism, dhRes, was metabolized almost completely, with only traces of the unchanged molecule being found. A sulphate conjugate was identified as the main metabolite of tRes in our model, while a glucuronide conjugate was the major metabolite of cRes and dhRes. Since metabolism of simple phenolics and polyphenols plays a crucial role in their bioavailability, detailed knowledge of their transformation is of high scientific value.

## 1. Introduction

Stilbenoids are a group of plant phytoalexins occurring in various edible and medicinal plants. Currently, more than 400 derivatives of stilbene have been identified [1]. Among these, the most well-known is *trans*-resveratrol, which is found in grapes (*Vitis vinifera* L.) and therefore red wine, peanuts (*Arachis hypiogaea* L.), and in a wide range of berries (genus *Vaccinium* L.) [2,3]. Resveratrol occurs in both *trans*- and *cis*- form; *trans*-resveratrol is believed to be the bioactive form [3]. *Trans-*resveratrol has been intensively studied due to its connection with the French paradox, where low mortality from coronary heart disease was observed despite high intake of saturated fat in a French sample population [4,5]. Up to now, *trans*-resveratrol has been tested in more than 160 clinical trials, connected mostly with the treatment of diabetes mellitus, cancer, cardiovascular and neurodegenerative diseases. *Trans-*resveratrol displays great pharmacological potential, but in parallel, very low bioavailability [6]. After oral administration of *trans-*resveratrol (25 mg), its plasma concentration was detected to be lower than 5 ng/mL, however, total concentration of resveratrol metabolites was as high as 491 ± 90 ng/mL [6,7].

Both *trans-* and *cis- r*esveratrol are intensively metabolized in the intestine and liver. Similar to most of the other polyphenols, resveratrol undergoes microbial metabolism in the colon [8]. Ex vivo studies have shown that resveratrol metabolites differ among individuals, and thus far three of its metabolites have been identified in an in vitro faecal fermentation system: dihydroresveratrol, 3,4′-dihydroxy-*trans*-stilbene, and 3,4′-dihydroxybibenzyl (lunularin) [8,9]. Resveratrol and its catabolites are absorbed by epithelial cells; once in the enterocyte, resveratrol is conjugated into glucuronides or sulphates and partly transported back to the intestinal lumen [10,11]. In vivo and in vitro, using a Caco-2 cell line model, some resveratrol metabolites have been identified including resveratrol-4′-*O*-glucuronide, resveratrol 3-*O*-glucuronide, resveratrol-3-*O*-sulphate, and resveratrol-4′-*O*-sulphate, in both *cis*- and *trans*- forms [11,12,13]. A part of resveratrol and its metabolites is pumped back to the lumen by efflux proteins, such as the ATP-binding cassette (ABC) transporters. Multidrug resistance-associated protein (MRP2) and breast cancer resistance protein (BCRP) were identified as two transporters involved in the efflux of resveratrol conjugates [10]. Transport of resveratrol into the vascular endothelial cells is either by passive diffusion or a sodium-dependent glucose transporter 1 (SGLT1)-mediated pathway [14].

The high bioactivity but low bioavailability of resveratrol is often referred to as the “Resveratrol paradox,” which has several potential explanations. A recent study [14] mentioned the crucial role of an intracellular resveratrol pool, which might be even more important than the serum level in vivo. It has also been suggested that metabolites of resveratrol might act as inactive reservoirs for resveratrol generation [15]. In vivo studies in mice [16] and in obese humans [17] showed a significant biological effect of resveratrol even at low plasma concentrations 10–120 ng/mL and 231 ng/mL, respectively. These studies showed that resveratrol administration increased mitochondrial function through the activation of SIRT1 (silent information regulator) and PGC-1α (peroxisome proliferator-activated receptor γ coactivator), which in mice translated into an increase in energy expenditure, improved anaerobic capacity, enhanced sensorimotor function, and, in humans, a reduction in sleep and in the resting metabolic rate. The conjugation also leads to higher water solubility of metabolites and their easier elimination from the organism by urine. It has been shown that after oral dose of sulphate and glucuronide conjugates the amount excreted in the urine accounts for around 24% and 13%, respectively [6].

Therefore, in recent years, studies have focused more on the biological activity of resveratrol metabolites. It has been shown that some resveratrol metabolites are more cytotoxic towards tumor cells (HT-29, Caco-2, and MCF-7) compared to parental resveratrol, and less toxic towards noncancerous HEK-239 cell lines [18,19]. Some metabolites of resveratrol were also shown to inhibit angiogenesis, and telomerase production [18]. Piceid, a glucuronidated metabolite of *trans-*resveratrol, exhibits greater scavenging activity against hydroxyl radicals than *trans-*resveratrol in vitro [20].

While the transport and metabolism of *trans*-resveratrol in an intestinal model is well known, sufficient data are lacking for its *cis*-isomer (present with tRes in wines, although at a lower abundance) and its main colon catabolite. This knowledge is very important to fully understand the biological activity of these compounds, as well as the bioactivity of stilbenoids in general. Therefore, we aimed to investigate the phase II metabolism of *trans*-resveratrol (tRes), *cis*-resveratrol (cRes), and dihydroresveratrol (dhRes) (Figure 1) in a standard model using a Caco-2 cell line in a Transwell^TM^ cellular system.

## 2. Materials and Methods

### 2.1. Preparation of Compounds

*Trans*-resveratrol (tRes) was obtained from Merck (Darmstadt, Germany), *cis*-resveratrol (cRes) and dihydroresveratrol (dhRes) were obtained from ChemFaces (Hubei, China), all in purity of 98%. Resveratrol-3-*O*-glucuronide and resveratrol-3-*O*-sulphate, both in purity of 95% were obtained from Cayman chemical company (Michigan, USA). All samples were diluted in DMSO, and then HBSS (Hanks’ Balanced Salt solution) at the day of experiment. The final concentration of 20 µM was based on previous MTT (3-(4,5-dimethylthiazol-2-yl)-2,5-diphenyltetrazolium bromide) cytotoxicity tests, published before [21]. The concentration of DMSO in the final solution did not exceeded 2% to assure no effect on Caco-2 monolayer.

### 2.2. Cell Cultures

The human epithelial intestinal cell line Caco-2 was obtained from American Type Tissue Collection (Rockville, MD, USA), and 25^th^ passage of Caco-2 cells was used in the present experiment. Conditions for growing and passaging of the cells were previously described here [21].

### 2.3. Permeability Assay

The permeability assay protocol was conducted according to [22].

#### 2.3.1. Preparation of Inserts with Caco-2 Cells

Cells were cleaned from the medium and re-suspended in DMEM (-F12, Dulbecco’s modified Eagle’s medium) supplemented with 10% FBS (fetal bovine serum), 1% non-essential amino acids, 1% penicillin and streptomycin, all obtained from Sigma-Aldrich (Prague, Czech Republic) at a concentration of 0.6 × 10^6^ cells/mL. The inserts in 24-well cell culture clusters were pre-wetted with 50 µL of medium for at least 2 min before cell seeding. The cells were applied to the apical side (Figure 2) in seeding density of 2.6 × 10^5^ cells/cm^2^. The basolateral chamber was filled with 1 mL of DMEM and incubated at 37 °C, 5% CO_2_-humidified atmosphere. To remove non-adherent cells the apical medium was removed after 6 h of incubation and replaced with 0.5 mL of fresh DMEM. The medium was changed daily, firstly aspirated from the basolateral and then from the apical side, fresh DMEM was added first to the apical and then to the basolateral side. Cells were grown for 21 to 25 days to create a fully confined monolayer. Last change of the medium was carried out 16 h before the experiment.

#### 2.3.2. Measuring of the Monolayer Integrity

The filter inserts with a monolayer of Caco-2 cell line were washed three times with HBSS pre-warmed to 37 °C and at pH 7.4. Transepithelial electrical resistance (TEER) was required to be at least 600 Ω, to ensure the integrity of the cellular barrier. Then, a Lucifer yellow dye at a concentration of 25 µM was added and the plates were incubated at 37 °C, and 5% CO_2_ atmosphere for 1 h while shaking (150 rpm). The plates were measured in a Tecan Infinite M200 reader (Excitation/Emission wavelength 480 nm/530 nm). Only the inserts with integrity higher than 95% were used.

#### 2.3.3. Metabolism and Absorption of Tested Compounds

Inserts were washed three times, and 500 µL of solution of parental compounds in a concentration of 20 µM was added to the apical side, 1000 µl of HBSS was added to the basolateral side. The samples from the apical side (50 µL) were taken immediately (time point 0 h). Plates with inserts were incubated on an orbital shaker (150 rpm) in a CO_2_ incubator (37 °C, 5% CO_2_-humidified atmosphere). Samples from the basolateral compartment were collected at time points 0.5, 1, 1.5, 2, 3, and 4 h, respectively. 500 µL of HBSS was removed and replaced with 500 µL of fresh HBSS. At the end, the samples from the apical side were collected and inserts were washed three times with HBSS. TEER was measured to make sure that the integrity of the cellular barrier was not broken (>500 Ω). To evaluate the intracellular contents of the tested compounds, the remaining cells on inserts were extracted with 100% methanol. All samples were stored at −80 °C until the analysis.

### 2.4. LC/MS Analysis

#### 2.4.1. Standards

Standards of tRes and dhRes were kept in dry form, with exception of cRes, which was provided as a solution in ethanol, and stored at −18 °C up to one year. Due to a relative instability of standards in solvent, especially tRes and cRes, fresh stock solutions were prepared before each measurement. Calibration samples and quality control (QC) samples were prepared by diluting of stock solutions in methanol/formic acid (99/1) to make calibration series in the range of 1–1000 ng/mL, and kept at 4 °C.

#### 2.4.2. Sample Purification

All samples from the permeability assay were centrifuged (5 min, 15,000× *g*; Rotanta 460R, Hettich, Germany). The samples from 0 h time point were diluted in 450 µL in methanol/formic acid (99/1) (1:9), while all other samples were diluted 1:1. Each sample was spiked with 20 µL of [^13^C_6_] *trans*-resveratrol solution in methanol (2 µg/mL) used as an internal standard.

#### 2.4.3. LC/MS Analysis of Metabolites

Analyses were performed on a LC/MS system consisting of a UHPLC chromatograph Ultimate 3000 Thermo Fisher Scientific (Sunnyvale, CA, USA) coupled with a quadrupole time-of-flight (Q-TOF) mass spectrometer with ultra-high resolution and a high mass accuracy (HRAM) Impact II (Bruker Daltonics, Bremen, Germany) equipped with an electrospray ionization (ESI) source. Chromatography was carried out on a Kinetex 1.7 μm F5 100 Å 100 × 2.1 mm column (Phenomenex, CA, USA). Detailed description of analysis was previously published by Jarosova et al., 2019 [8]. The list of all searched and detected compounds is shown in Appendix A, Table A1. After each five sample injections the QC (50 ng/mL of each analyte in the mixture) injection was performed. The validation parameters are shown in Appendix B, Table A2. Briefly, accuracy of the LC/MS measurement was calculated from repeated injections of standard solution and was in the range of 0.12–3.15% RSD and limit of detection calculated as signal to noise ratio 3:1 was in the range of 7.1–17.4 ng/mL. For metabolites, where analytical standards were not available, abundance was expressed as intensity, referring to the peak area.

### 2.5. Statistical Analysis

Because of reduction in donor concentration on the basolateral side after every sampling, the actual concentrations at each time point were counted according to following equation:
$$C_{A}=\frac{\Sigma C_{P}}{2}+C_{M}$$
where C_A_ is the actual concentration at the time point, C_P_ are the previous concentrations, and C_M_ is the concentration measured at the time point. Values are expressed as a mean ± standard deviation. Microsoft Excel, SPSS (IBM corp., Armonk, NY, USA) version 25, and Statistica12 (StatSoft, Tulsa, OK, USA) were used for basic statistical analysis and graph creation. Quantitative data were normalized to 20 µM to correct the minor dilution errors. The experiments for tRes and cRes were carried out in four biological replicates, dhRes in five biological replicates. Each of them was prepared in three technical repetitions.

## 3. Results

The fate of tRes, cRes, and dhRes in the intestinal model significantly differed for each test compound (Figure 3). From the initial 20 µM, tRes was mostly transported through the membrane to the basolateral side (57.2 ± 2.9%) while 22.1 ± 4.5% either remained or was pumped back to the apical side, and 20.3 ± 7.2% was transformed or metabolized. Its isomer, cRes, was transformed or metabolized by 62.1 ± 2.4%; 32.2 ± 1.7% was transported unchanged to the basolateral side; and only 5.5 ± 2.2% was detected on the apical side at the end of the experiment. On the contrary, only traces of their metabolite, dhRes, were detected on the apical or basolateral side after 4 h of incubation. Most dhRes, 99.4 ± 0.06%, was metabolized or transformed by Caco-2 cells, and only 0.6 ± 0.3% was detected unchanged on the basolateral side and 0.01 ± 0.01% on the apical side. Less than 0.5% of all the parent compounds (0.4 ± 0.1% for tRes, 0.1 ± 0.0% for cRes, dhRes not detected), accumulated in the cells.

As seen in Figure 4, three metabolites of tRes were detected and identified as tRes-sulphate and two tRes-glucuronides. Compared to the standards, these metabolites were identified as tRes-3-*O*-sulphate, tRes-3-*O*-glucuronide, and tRes-4′-*O*-glucuronide. Sulphate was the dominant metabolite of tRes. After 4 h of experiment, 3.96 ± 0.84 µM of tRes-3-*O*-sulphate was detected on the basolateral side, 2.75 ± 0.53 µM on the apical side, and 0.09 ± 0.02 µM in the cells, respectively. Metabolites tRes-3-*O*-glucuronide, and tRes-4′-*O*-glucuronide were detected on the basolateral side at 1.15 ± 0.15 µM and 0.39 ± 0.06 µM, respectively, and on the apical side at 0.52 ± 0.09 µM and 0.18 ± 0.05 µM, respectively. Similarly, three metabolites, cRes-sulphate and two cRes-glucuronides, were detected for cRes. However, contrary to tRes, cRes-glucuronide seemed to be the dominant metabolite of cRes. Two metabolites, dhRes-sulphate and dominant dhRes-glucuronide, were detected for dhRes. No isomeric transformations were detected for any compound during the incubation as well as during the storage of the samples.

The time-dependent changes of metabolism are shown in Figure 5. The concentration of tRes on the basolateral side increased continuously and slowed down after 3 h of incubation. On the contrary, the increment in concentration of its main metabolite, tRes-sulphate, sped up after 2 h of incubation. The concentration of both tRes-glucuronides rose slowly but continuously, during the entire incubation. Contrary to the pattern observed for tRes, the intensity of cRes on the basolateral side increased rapidly during the first hour of incubation and reached a plateau after 2 h of incubation. Regarding the three metabolites of cRes, they all continuously increased in intensity during the entire incubation. Only traces of unchanged dhRes were found on the basolateral side, and both of its metabolites slowed down in their intensity increase after 2 h of incubation.

## 4. Discussion

The metabolism of natural compounds before they reach the bloodstream plays a crucial role in their bioactivity. When reaching the colon, stilbenoids are intensively metabolized by colon microbiota and transported through the enterocytes, possibly mainly in the form of conjugates. The aim of our experiment was to find the most important metabolites and compare the metabolic fate of tRes, cRes and dhRes in the intestinal epithelium using an epithelial Caco-2 cell line model. We found major differences among each of the parent compounds.

The majority of tRes (57.2%) was transported unchanged to the basolateral side; this was shown in a similar study [23], where after 4 h of incubation, 53% of unchanged tRes appeared on the basolateral side. This indicates that metabolic degradation of tRes during the intestinal absorption may not be an important factor influencing its bioavailability. However, only trace amounts of unchanged tRes were detected in vivo in plasma after oral administration, as shown in studies [6,7]. As described in a previous study [8], tRes is metabolized by gut microbiota, at different intensities per individual donors (77–11% of unchanged tRes appeared after 48 h of incubation), which together with intensive metabolism in the liver might contribute to its low plasma concentration in vivo. Another factor responsible for tRes low bioavailability could be its bidirectional transport through epithelial cells. After 4 h of incubation, 22.1% of unchanged tRes was found in the apical chamber. Interestingly, only traces of tRes have been detected intracellularly after 4 h. This could be explained by intensive active transport of tRes, which was previously observed and MRP2 was identified as the responsible apical efflux transporter [24]. On the other hand, another study [25] showed completely different results by detecting high intracellular concentrations of tRes. However, the analysis of the cells was conducted after one hour of incubation, which might have, together with a slightly different extraction method, caused the divergence from our results, which were obtained after 4 h. In our study, 20.3% of tRes was metabolized or differently transformed, and three metabolites of tRes, dominant tRes-3-*O*-sulphate, tRes-3-*O*-glucuronide, and tRes-4′-*O*-glucuronide, were detected. The intensity of tRes-3-*O*-sulphate was about three times higher than the sum of both glucuronides. A similar study [25] of tRes detected two metabolites, tRes-monoglucuronide and tRes-monosulphate, with a trend similar to the one observed by us. In a different study using a rat small intestine model [26], a glucuronide conjugate of tRes was detected as a major metabolite. An in vivo study in pigs also showed a tRes-glucuronide as a main metabolite in fluids and organs [27]. This might be caused by interspecies differences of intestinal conjugation enzymes. The concentrations of all metabolites grew during the entire incubation, and after 4 h they were also detected on the apical side of inserts. No *cis* isomers or hydrogenated metabolites were found during the passage of tRes through Caco-2 cells.

The metabolism of cRes differed significantly from its *trans* isomer. The majority of cRes was metabolized and only 32.2% passed unchanged to the basolateral side. Only 5.5% of cRes was detected on the apical side after 4 h of incubation, indicating bidirectional transport with strong predominance from the apical to the basolateral side, which is an important factor affecting its bioavailability. Similar to tRes, only traces of cRes were detected intracellularly after 4 h of incubation, indicating the efficient active transport of these compounds out of the cells. Three metabolites of cRes were detected, cRes-sulphate, and two cRes-glucuronides. In contrast to tRes, which was mainly conjugated with a sulphate, cRes was shown to conjugate mainly with glucuronic acid. The intensity of cRes glucuronides was about seven times higher than that of sulphate. The glucuronidation of tRes and cRes by Caco-2 cell lines was demonstrated in an earlier study [28] where the rate of cRes glucuronidation was up to 90-fold higher than that of tRes. Similar to tRes, no *trans* isomers or hydrogenated metabolites were found in our model.

Dihydroresveratrol is the main gut microbiota metabolite of resveratrol as was observed in our previous study [8]. After application on Caco-2 cells, the vast majority (99.4%) of dhRes was conjugated and only traces of unchanged dhRes were found on both the apical and basolateral side of the Caco-2 cells. Two metabolites of dhRes were detected in our model, dhRes-sulphate and dominant dhRes-glucuronide. The intensity of dhRes-glucuronide was about four times higher than that of dhRes-sulphate. As mentioned earlier, in the Caco-2 cells cRes forms glucuronides at a higher rate than tRes. In the gastrointestinal tract, UDP-glucuronosyltransferases (UGT) are active in glucuronidation of tRes or cRes, and it has been shown that UGT has a greater substrate specificity towards cRes than to tRes [28]. The presence of a single bond in a dhRes molecule allows configuration changes that can make it more similar to either the *cis* or *trans* isomer of resveratrol. This bond arrangement allows dhRes to comply with different active site positions of UGT present in cells and it might explain its prevalent glucuronidation similar to cRes. Interestingly, dhRes-sulphate was the only metabolite detected at a higher intensity (2-fold) on the apical side than on the basolateral side and only traces of it were detected intracellularly. This indicates efficient active transport of this metabolite to the apical side, which may decrease its absorption into the blood stream and simultaneously prohibit the potentially positive effect within enterocytes. In an in vivo study in rats [29], 30 min after oral administration of 60 mg/kg of dhRes, a glucuronide conjugate was most abundant in plasma (33.5 µM), and a sulphate conjugate was also present at lower intensities (6.4 µM). Unchanged dhRes was also detected in plasma 30 min after the oral administration at very low intensities (0.88 µM). During the passage of dhRes through the Caco-2 cells no dehydrogenated analogues were found, showing that dhRes is not a source of resveratrol.

## 5. Conclusions

In conclusion, the permeability of tRes, cRes and dhRes in a Transwell^TM^ system using Caco-2 cell lines was explored, detecting altogether eight principal metabolites. Our results showed significant differences in the metabolism of resveratrol configurational isomers. The compounds differed in degrees of metabolism, tRes was metabolized by 20%, followed by cRes (62%) and dhRes (99%) A conjugate with sulphate was identified as the main metabolite of tRes, while a glucuronide was a major metabolite of cRes and of dhRes.

## Figures and Tables

**Figure 1 nutrients-12-00595-f001:**
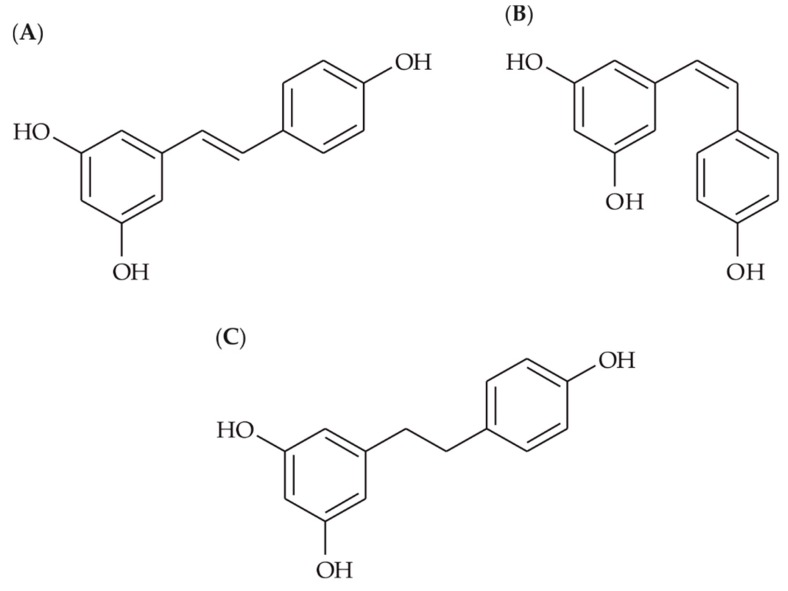
Structures of parent compounds (**A**) *trans*-resveratrol, (**B**) *cis*-resveratrol, (**C**) dihydroresveratrol.

**Figure 2 nutrients-12-00595-f002:**
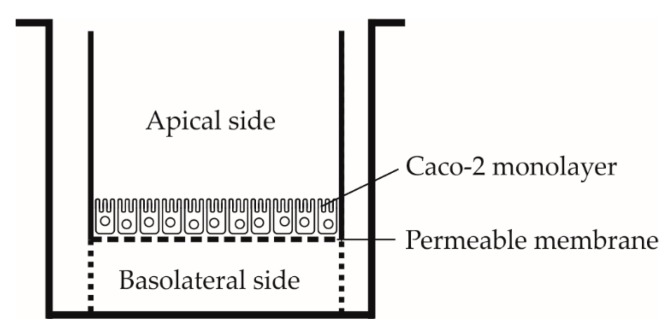
A schema of Transwell^TM^ cellular system.

**Figure 3 nutrients-12-00595-f003:**
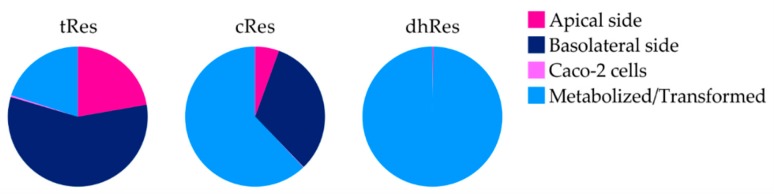
Fate of parent compounds in Transwell^TM^ cellular system (% mol).

**Figure 4 nutrients-12-00595-f004:**
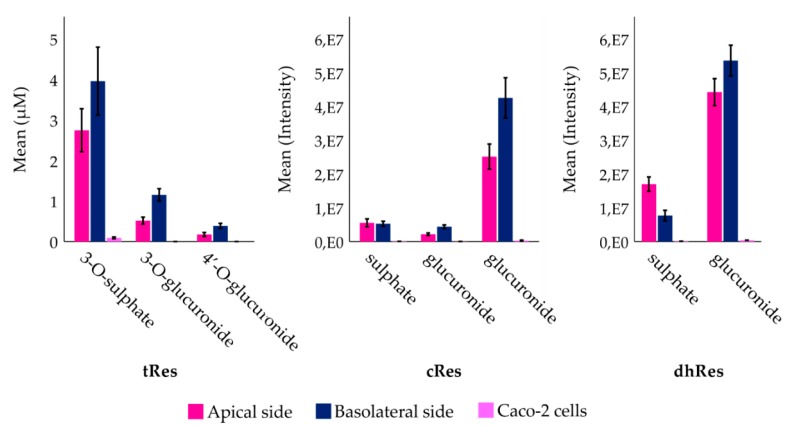
Metabolites observed after 4 h of incubation in Transwell^TM^ cellular system. Values obtained from LC/MS for tRes are expressed as mean concentration ± standard deviation, *n* = 4; for cRes and dhRes values are expressed as mean intensity ± standard deviation, *n* = 4 and *n* = 5, respectively. Steric positions of bonded conjugated units on cRes and dhRes cannot be specified, due to lack of confirmed standards.

**Figure 5 nutrients-12-00595-f005:**
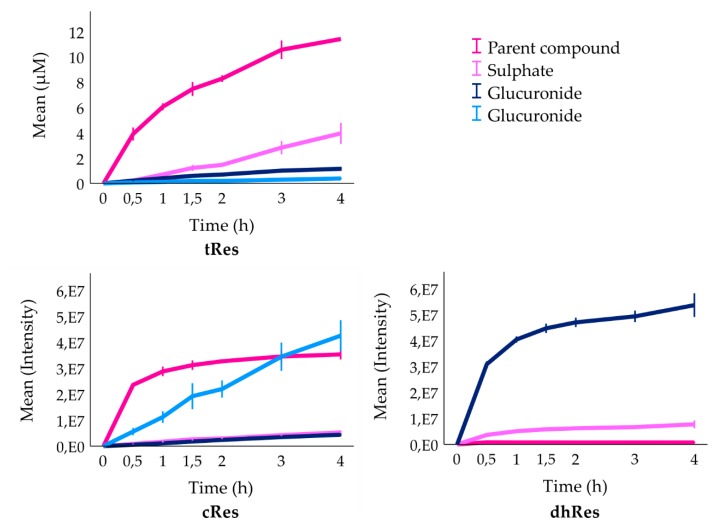
The changes of parent compounds and their metabolites on the basolateral side of inserts in Transwell^TM^ cellular system. Values obtained from LC/MS for tRes are expressed as mean concentration ± standard deviation, *n* = 4; for cRes and dhRes values are expressed as mean intensity ± standard deviation, *n* = 4 and *n* = 5, respectively (see Materials and methods).

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
