# Peer review of "Metabolism of cis- and trans-Resveratrol and Dihydroresveratrol in an Intestinal Epithelial Model"

_nutrients, 2020, doi:10.3390/nu12030595_

Round 1
Reviewer 1 Report
In this study, the authors investigated the transport and metabolism of cis- and trans-resveratrol and dihydroresveratrol using Caco-2 model cell line. Eight different metabolites were detected and characterized by LC-MS, consisting of glucuronides and sulphates of parent compounds. The three compounds showed differential transport and metabolism patterns. Conjugate with sulfate appeared to be the main metabolite of trans-resveratrol while glucuronides were determined as the major metabolites of cis-resveratrol and dihydroresveratrol.
Main critiques:
The “Error! Reference source not found” message can be found throughout the manuscript. Such mistakes should be corrected before submission. All figures were not referred in the manuscript. They should be mentioned in the result and/or discussion. It could be better to directly use the identities of characterized metabolites, not their codes such as “M1-S”, “M2-G” etc. How many repeats were carried out in each experiment? Information was missing. In figures 4 and 5, different units were used between tRes and other two compounds. This is understandable since not all standards are available. However, it would be better to also quantify the metabolites in cRes and dhRes groups using the standard of tRes metabolites. They should have similar MS response. Data presented in ion intensity have less meaning. Transform between the two geometric resveratrol isomers was not detected. This is very important observation and should be included in the result, not only in the discussion.Author Response
Dear reviewer, please find our rebuttal letter in the attachment.

Reviewer 2 Report
Well-written, concise, well-documented manuscript. One minor typo noted on Line 183: replace 0,39 with 0.39. Otherwise, ready for publication w/ no major edits.
Author Response
Dear reviewer,
Thank you, please find our rebuttal letter to other reviews in the attachment.

Reviewer 3 Report
The paper is about metabolism of cis and trans Resveratrol and dihydro-resveratrol in an intestinal model using Caco-2-cells. The work is interesting and give us new knowledge about the metabolism of this bioactive compound. However, there are some critical points that need to be clarified or corrected:
Line 39, 44, 45, 49 I think that will be convenient to put a reference.
Line 70 Please, when an acronym appears for first time you should explain the meaning
Line 85, 110, 166, 178, 195 you have an error in the reference or something similar, please correct it.
Line 96 Please, explain the meaning of MTT
Line 138 How long time did the authors store the standards? it will be interesting to know.
2.4.3 section It is necessary to know LQ (quantification limit) and LD (detection limit) limits, and give it in this section. It would be better if you have validated the method. If you did it, it would be better give us the validation parameters. Did the authors used quality control in the analysis method?
Line 151 I think that it would be politer to write ‘previously reported by Jarosova et al. 2019’ instead ‘here’. Please change it and check it in entire document.
Line 156 I think it deserves a reference.
Line 175, Figure 3 did you mean in the intracellular samples? Please be more specific.
Line 179 did the author identify the sulfate derivative with a standard? It could be interesting to know it.
Figure 4 why did not the authors would quantify cRes and dhRes? It would be better expressed all the results in the same units.
Line 198 I don’t understand “both rose glucuronides” I think there are blue.
Line 199, 201 you cannot talk about cRes and dhRes concentration you should talk about content or intensity. Please rewrite this sentence, I don’t understand.
Line 220 Have the authors evaluated if there is first-step effect? could be an explanation for low plasma concentration.
Line 228 to 240 to compare with other studies it is necessary to know if the method of analysis was similar and if both papers used similar intracellular extraction methods. The compounds detected depends of sensibility of the analysis method.
Line 253 I think it would be better to use produced or released instead delivered.
Line 261 I don´t understand the word imitation in this sentence.
Line 262 to 272 could the authors consider the production of sulfate compounds is a mechanism to eliminate metabolites from the organism, being that the sulfonation increase the solubility making easily the elimination? Please look for references.
Author Response
Dear Reviewer,
Please find our rebuttal letter in the attachment.
